# Barriers and Enablers for Equitable Healthy Food Access in Baltimore Carryout Restaurants: A Qualitative Study in Healthy Food Priority Areas

**DOI:** 10.3390/nu16173028

**Published:** 2024-09-08

**Authors:** Shuxian Hua, Vicky Vong, Audrey E. Thomas, Yeeli Mui, Lisa Poirier

**Affiliations:** 1Department of International Health, Johns Hopkins Bloomberg School of Public Health, Baltimore, MD 21205, USA; shua8@jhmi.edu (S.H.);; 2Department of Health Policy and Management, Johns Hopkins Bloomberg School of Public Health, Baltimore, MD 21205, USA; vvong1@jhmi.edu; 3Department of Health, Behavior, and Society, Johns Hopkins Bloomberg School of Public Health, Baltimore, MD 21205, USA

**Keywords:** independently owned restaurants, healthy food access, food desert, food policy, nutrition program

## Abstract

Black neighborhoods in the U.S., historically subjected to redlining, face inequitable access to resources necessary for health, including healthy food options. This study aims to identify the enablers and barriers to promoting equitable healthy food access in small, independently owned carryout restaurants in under-resourced neighborhoods to address health disparities. Thirteen in-depth interviews were conducted with restaurant owners in purposively sampled neighborhoods within Healthy Food Priority Areas (HFPAs) from March to August 2023. The qualitative data were analyzed using inductive coding and thematic analysis with Taguette software (Version 1.4.1). Four key thematic domains emerged: interpersonal, sociocultural, business, and policy drivers. Owners expressed mixed perspectives on customers’ preferences for healthy food, with some perceiving a community desire for healthier options, while others did not. Owners’ care for the community and their multicultural backgrounds were identified as potential enablers for tailoring culturally diverse menus to meet the dietary needs and preferences of their clientele. Conversely, profit motives and cost-related considerations were identified as barriers to purchasing and promoting healthy food. Additionally, owners voiced concerns about taxation, policy and regulation, information access challenges, and investment disparities affecting small business operations in HFPAs. Small restaurant businesses in under-resourced neighborhoods face both opportunities and challenges in enhancing community health and well-being. Interventions and policies should be culturally sensitive, provide funding, and offer clearer guidance to help these businesses overcome barriers and access resources needed for an equitable, healthy food environment.

## 1. Introduction

Over the past 50 years, Americans have significantly increased their consumption of food prepared outside the home, constituting 34% of the total household calories consumed [1]. In 2022, spending on food away from home in the U.S. was 16% higher than the previous year [2], contributing more than 570 additional calories per day to the American diet, on average [3]. While these figures encompass all meals prepared outside the home, particular attention has been directed towards those prepared by fast-food services such as carryouts (i.e., small, independently owned restaurants with primarily pickup services and limited seating), which provide mostly energy-dense, nutrient-poor options [4].

Despite overall improvements in the nutritional quality of fast-food meals in the U.S. from 2003 to 2016, studies have shown that they still do not match the dietary quality of full-service restaurants [5]. Moreover, disparities in access to healthy prepared foods are evident at the neighborhood level, with a disproportionate concentration of fast-food restaurants located in predominantly low-income, African American census tracts, while non-Hispanic White neighborhoods have a higher prevalence of non-fast-food restaurants overall [6]. These inequities in healthy food access are manifestations of ‘structural racism’, defined as macro-level conditions (e.g., laws, institutional policies, and entrenched norms) that restrict the opportunities, resources, power, and well-being of individuals and populations based on race/ethnicity [7]. Consequently, experts in nutrition, food security, and food systems are increasingly exploring the role of structural drivers in shaping disparities in access to healthfully prepared food [8].

Efforts to address these disparities have included a variety of intervention strategies ranging from expanding menus with healthy options, using financial incentives and educational campaigns to promote healthy choices, reducing portion sizes, and offering healthier substitutes [9,10,11]. A systematic comparison of 27 interventions conducted in both rural and urban areas throughout the U.S. revealed that the only interventions demonstrating sufficient evidence of effectiveness involved point-of-purchase information combined with increased availability of healthy menu items, underscoring the significance of healthier food procurement in restaurants for increasing healthy food access [12]. A scoping review that examined the motivations and obstacles for procuring healthy foods faced by corporate-owned establishments compared to independently owned restaurants found that the latter demonstrated a stronger commitment to the well-being of their customers and local communities but had constraints related to available financial resources [13].

Despite the extensive body of research on restaurant interventions, there is limited knowledge regarding the structural factors influencing healthy food access in carryout restaurants. Previous studies have shown that business decisions and interpersonal dynamics shape menu offerings. Carryout restaurant owners have reported receiving negative feedback from customers when offering healthy choices and expressed concerns that such offerings might have a detrimental impact on their business [14]. Other research has highlighted concerns around low customer demand for healthy foods, potential revenue loss, and non-financial challenges related to sourcing healthy ingredients [15]. A closer examination of these concerns which decode policies shaping the neighborhood food environment, offers a promising path forward for establishing a healthier and more equitable carryout food system. To identify key areas for implementing interventions or policies designed to improve access to healthy food in carryout restaurants, this paper seeks to answer the following questions: What are the barriers and enablers for promoting equitable access to healthy food in Baltimore’s carryout food system at the interpersonal, sociocultural, and business levels? Additionally, what are the policy drivers influencing small business operations and their provision of healthy foods?

## 2. Materials and Methods

### 2.1. Study Context

In Baltimore, Maryland, equitable access to healthcare resources and healthy food is intricately linked to historical and social factors, particularly structural racism and redlining [16]. The urban food environment has been significantly influenced by racial and economic segregation, which began to take shape in the early 20th century with the implementation of a racial housing covenant in 1910, restricting African Americans and other racial minorities from buying or occupying homes in neighborhoods designated as “white only” [17]. Such covenants, alongside other forms of structural racism in housing, have led to substantial disparities in access to resources, including but not limited to quality education, employment, food and healthcare [7]. As of 2018, Baltimore City had a population of approximately 600,000, with 31% of black residents living in an HFPA, compared to only 9% of White residents live in Priority Areas [18]. To be considered as a HFPA, an area must have the lowest-tier Healthy Food Availability Index (HFAI), a score based on the presence of basic food staples and healthy food options ranging from 0 to 28.5. Additionally, the median household income must be at or below 185% of the federal poverty level, over 30% of households must not have a vehicle, and the distance to the supermarket must be greater than a quarter of a mile [18].

Meanwhile, the immigrant population in Baltimore City has been rapidly growing, with an 11.5% increase from 2010 to 2021 [19]. Today, immigrants constitute only 8% of the total population, yet they own 21% of the city’s businesses, notably investing in neighborhood enterprises such as corner stores and carryout restaurants [20,21,22]. However, challenges such as language barriers and undocumented status often lead to their underrepresentation in research studies, leaving their needs and concerns underserved [23].

### 2.2. Sampling and Participant Recruitment

We used purposive and random sampling for recruitment. Zip codes were purposively selected based on fulfilling the criteria of an HFPA (as defined in Section 2.1) by the city government. These criteria were chosen to focus on carryout restaurants most likely experiencing inequitable access to healthy food. Demographic data were obtained from the 2020 Census. To ensure geographic diversity, zip codes were purposively selected to balance representation, aiming for roughly equal representation from east and west Baltimore. In total, 10 zip codes were selected.

Carryout restaurants were eligible if they met the following criteria: (1) the owner was 18 years or older and (2) the carryout restaurant was located in Baltimore City. A list of carryout restaurants in each selected zip code was obtained from November 2022 data provided by the Maryland Department of Health. These restaurants were randomized using a random number generator to create the order in which research assistants would attempt recruitment. The goal was to recruit at least one restaurant per zip code. Additionally, specific restaurants were also recommended by the project’s Community Advisory Board (CAB) comprised of a carryout restaurant owner and food access planners from the Baltimore City Department of Planning. The intention was for the recruitment phase to continue until at least one restaurant from each zip code was successfully recruited into the study or until data saturation was reached, as determined collectively by the research team. Data saturation was determined when no new themes or information emerged from the subsequent interviews [24]. Ultimately, the team was able to recruit at least one restaurant from eight of the ten zip codes. The research team approached 190 restaurants and successfully interviewed 13 restaurant owners or managers. Roughly 33% (66) of the restaurants approached were permanently closed, the owner was not available at 22% (42), 13% (26) did not meet inclusion criteria, 12% (22) were closed repeatedly or were unable to be located, 11% (21) did not want to participate, and 1.5% (3) were interested, but were lost to follow-up when trying to schedule an interview.

### 2.3. Data Collection

From March to August 2023, thirteen interviews were conducted with carryout restaurant owners by four trained research assistants. These interviews were semi-structured and based on an in-depth interview (IDI) guide developed by the research team, which included the following 5 sections of questions: general information, carryout operations, food shopping and sourcing, relationships with the community, and food policy (Appendix A). Interviews lasted between 30 and 60 min and were conducted in-person, over the phone, or via Zoom and recorded using the voice memos application, with prior consent obtained from all participants. Participants received a USD 25 gift card as compensation.

### 2.4. Data Management and Analysis

Audio files of interviews were transcribed using Scribie’s cloud-based transcription service as of 8 April 2016 [25]. Cleaned transcripts were uploaded into Taguette software (Version 1.4.1) for qualitative data analysis [26]. A codebook was developed inductively through group discussion. Three research assistants independently coded half of the transcripts, then met with the research team to discuss the codebook’s applicability to the transcripts overall. Following several rounds of discussion, the revised codebook was used to code the remaining transcripts. Thematic analysis was used to analyze qualitative data. Codes were organized into themes encompassing interpersonal, sociocultural, business, and policy drivers with sub-themes identified. All themes were reviewed in collaboration with CAB members on 21 August 2023.

## 3. Results

Thirteen carryouts from eight zip codes participated in the study. Table 1 presented their descriptive characteristics. Of these, six carryouts were inherited from previous generations, while seven were established by new entrepreneurs. Regarding their offerings, twenty-three percent of the carryouts offer Italian cuisine such as pasta and pizza, with an equal percentage serving seafood. Fifteen percent offer general American items like breakfast sandwiches and hot dogs, and another fifteen percent are stores specializing in fried chicken. Ethnic cuisines, such as Soul food (which originated in the American South and has significance to African American culture), Caribbean, and Asian food each make up eight percent. Thirty-eight percent are in neighborhoods with the lowest (<USD 750,000) small business investment.

### 3.1. Interpersonal Drivers

#### 3.1.1. Perceptions of Consumer Preferences

Owners expressed mixed perceptions about customers’ preferences for healthier food. Some owners observed that customers in the city, who have less access to high-quality food establishments compared to those in the surrounding county, demonstrate a preference for healthy options. One owner positioned this reality against the common perception that city customers merely dislike healthy food and emphasized that they should be offered healthy choices like those in more affluent areas (Table 2, quote 1). Aligned with this perception of healthy food preferences, owners also observed the growing demand for vegetables and vegetarian options and are accommodating these requests (Table 2, quote 2).

Conversely, some owners believed that city consumers refrain from making healthy food choices due to personal preferences, as well as other factors such as budgetary constraints and communication challenges. This perception contributed to owners’ reluctance to introduce healthier menu items due to anticipated low sales. A hot dog vendor in a food market, who admitted to “having no idea how to make that healthy” (Table 2, quote 3), speculated that Baltimore customers prioritize budget considerations when making their food choices, leading them to opt for cheaper fast-food options (Table 2, quote 4). Another breakfast carryout owner perceived that customers in less-resourced areas prefer the food they are accustomed to, suggesting it would be challenging to acclimate them to different options, including healthy ones (Table 2, quote 5). The belief that customers have unchangeable dietary habits was common among owners with limited English proficiency who spoke with native-speaking customers, as communication challenges contributed to misunderstandings of consumers’ preferences (Table 2, quote 6).

#### 3.1.2. Care for Consumer Well-Being

Owners showcased different ways of demonstrating care for their customers and the community. Some owners upgraded the healthfulness and affordability of the menus to make healthy food more accessible to those who previously had limited access to such options (Table 2, quote 7). They also catered to customers with dietary restrictions. Recognizing that consuming certain meats is taboo for people practicing specific diets or religions, owners introduced alternative protein options, such as poultry (Table 2, quote 8). Moreover, some owners demonstrated care for the community by expressing a commitment to serving food to anyone who comes to them in need, regardless of their ability to pay or physical disabilities, thus challenging social stigmas associated with marginalization (Table 2, quote 9). For owners, the motivation to foster a positive relationship with the community, treating every customer equally regardless of their socioeconomic status, is seen as advantageous for sustaining the business in the long run (Table 2, quote 10).

#### 3.1.3. Maximizing Transactions

A smaller number of owners discussed the transactional nature of running a food business, expressing a desire to maximize profits from their customers without offering much in return. One seafood carryout owner explained how the carryout was initially established by the original owner, who viewed it as a property investment for profit (Table 2, quote 12). Another owner, despite operating their establishment downtown with significant daily customer traffic from tourists and local workers, opted to live outside of Baltimore City due to its “bad reputation” (Table 2, quote 13). This profit-oriented mindset influenced their decisions on the types of food to offer, diverging from a focus on customer and community well-being to prioritizing the market viability of the food. As mentioned by the same owner who chose to live outside the city, the decision to introduce vegan options came after relocating to an area that attracted a different customer demographic (Table 2, quote 14).

### 3.2. Sociocultural Drivers

#### Multicultural Ownership Shapes Food Offerings

The cultural aspect of food offerings emerged as a recurring theme. Alongside commonly found fast-food-style items like fried chicken, sandwiches, pizza, and subs, many owners incorporated menu items reflecting their own cultural backgrounds (Table 3, quotes 1–2). For instance, a Black-owned carryout shared their motivation for offering “Soul food,” a term they used to denote African American cuisine, to provide novel options in a predominantly African American community already saturated with other types of carryouts (Table 3, quote 3). A few owners, originating from countries with food cultures distinct from those of the U.S., sought to blend home flavors and traditional dishes into their Americanized menus. This gradual introduction aims to acquaint customers with “something solid” (Table 3, quote 4) beyond the familiar “finger food”, as described by a Caribbean-origin owner (Table 3, quote 5). Similarly, an Afghan owner of a fried chicken carryout described introducing a grilled meat and rice dish to improve the healthfulness of the otherwise calorie-dense menu. The dish, containing multiple food groups (grains, vegetables, and dairy), not only enhances the nutritional content of the carryout’s menu but also provides the owner with an opportunity to share his cultural roots with his customers (Table 3, quote 6). This integration of different food cultures potentially enriches the diversity of the local food environment and may open up opportunities to provide customers with healthier food options.

### 3.3. Business Drivers

#### Cost of Business Impacts Carryouts’ Ability to Offer Healthy Options

Cost was a consistent theme occurring throughout the interviews, including both food costs and operational costs to keep the business running, such as rent, utility fees, taxes, and employee salaries. When discussing food costs, owners shared that they based their food purchasing decisions on price differences among suppliers. For small businesses like carryouts, adopting this cost-effective strategy is crucial for survival in a challenging macroeconomic climate (Table 4, quote 1). One owner highlighted the challenge of food price inflation over time, explaining that escalating food prices complicate the procurement of a diverse range of ingredients to meet customers’ varying preferences and dietary needs (Table 4, quotes 2–3). Beyond the challenges of sourcing cost-effective ingredients from suppliers, owners were also concerned about the additional costs associated with expanding their menu offerings. One owner of a pizza and sub carryout had plans to introduce a chicken dish would require “not too much oil” and would be “easy for older people to eat.” However, he mentioned that the specific equipment needed to cook this dish is expensive, causing him to delay the introduction of the new item to build up the necessary cooking capacity across months (Table 4, quotes 4–5). Ultimately, the increase in these costs will be reflected in the food prices charged to customers (Table 4, quote 6).

### 3.4. Policy Drivers

#### 3.4.1. Taxation

Many carryout owners voiced concerns over the increasing Maryland sales and business taxes, fearing that their business cannot “survive” due to rising costs and inflation (Table 5, quote 1). While specific taxation such as business tax and sales tax were a common complaint for the carryout owners, some noted that the taxation also negatively affected the consumers. Taxation dynamics can directly influence profitability, and if not properly communicated or understood, customers might mistakenly believe that the business is overcharging (Table 5, quote 2). A carryout owner was blamed by their customers, facing multiple complaints regarding Maryland’s sales tax affecting the cost of food items on the menu.

#### 3.4.2. Government Regulations and Policy

Maryland’s state-level policies often necessitate swift and sometimes costly changes to business practices, challenging carryout owners to adapt while maintaining service quality. Several carryout owners shared that the Health Department and other enforcers of general policy regulations have hindered or prevented business operations due to their inadequate and untimely delivery of information. These miscommunications range from information about permits, health inspections, and supply ordinances (Table 5, quotes 3–8).

Carryout owners described the city’s various inspections as a “challenge” since they had to cease operation waiting for repeated inspections from different government entities such as the Fire Department, and permit approval without a concrete timeline (Table 5, quotes 6–7). Furthermore, some carryout owners cited difficulty in scheduling these appointments (Table 5, quotes 8). This oversight is even present when carryout owners want to change or add menu items (Table 5, quote 9). Additionally, one carryout owner was concerned over increasing supply costs due to following the 2019 Baltimore City Foam Ban [27], an ordinance to support environmental sustainability efforts by banning foam to-go boxes in favor of compostable options (Table 5, quote 5).

When carryout owners were asked how to improve policies that would benefit their business operations, most carryout owners advocated for additional funding due to the variety of daily challenges that their small businesses face compared to corporate chains (Table 5, quote 10).

#### 3.4.3. Information Challenges

Some carryout owners reported difficulties in accessing beneficial information and resources regarding grant applications, business loans, and participating in the Supplemental Nutrition Assistance Program (SNAP) or Restaurant Meals Program (RMP), which may benefit their business and communities. SNAP is a federal program that provides low- or no-income individuals and families with food purchasing benefits to supplement the cost of groceries to ensure adequate health and nutrition [28]. Meanwhile, RMP is a sub-program of SNAP, operating as a state-optional program that allows certain SNAP beneficiaries, who are unable to prepare or store food for themselves, to buy prepared meals at restaurants with their SNAP benefits [29]. In recent years, the Maryland Department of Human Services has tried to support small businesses by providing grants, loans, and federal benefits opportunities, but carryout owners expressed that they were unaware that such programs existed. Those that did know explained that the application process and information was “not easy” and requires one to “really have to do your own groundwork” to seek information, especially for grants made specifically for minority small business owners (Table 5, quote 11). One carryout owner explained his SNAP application process, but was perplexed by the SNAP program’s initiative and navigation regarding the menu item requirements (Table 5, quote 12).

Factoring in foreign-born carryout owners, accessing resources to apply for and obtain licenses presents a bigger challenge due to language barriers or undocumented status. One carryout owner shared that they felt treated differently during the business ownership process compared to American owners due to their immigration status, having to wait 5 months for their tax and business identification information (Table 5, quote 13). Another owner expressed the need to travel to another state to attend classes for permits in their language (Table 5, quote 14).

#### 3.4.4. Uneven Investment across Neighborhoods

Investment disparities persist in Baltimore, with specific zip codes and neighborhoods with low incomes receiving minimal attention from policymakers. This presents an ongoing challenge that many carryout owners face. Existing establishments are less likely to be improved and supported by the local government as owners believe that the city’s politics and leaders influence the investments or disinvestments and allocation of funds (Table 5, quotes 15–17). One carryout owner blames and mistrusts their policymakers, “Once they get their juice, that’s all they care about”. Other carryout owners echoed this strained relationship with city officials, believing there is a lack of support for their businesses resulting in high turnover rates and inconsistent investments (Table 5, quote 16).

In addition, many carryout owners described how these conditions resonate throughout the community and culminate in crime, worse living conditions, inequitable food access, and declining business longevity (Table 5, quotes 17–18). One carryout owner criticized other carryouts, stating that they allow crime to occur in their vicinities to increase foot traffic (Table 5, quote 19). Although these were the issues highlighted, many carryout owners carried the sentiment that there needs to have more financial support for small business owners. One carryout owner indicated that there were specific areas within the city that need more funding from the government (Table 5, quote 17).

## 4. Discussion

In this study, we identified barriers, enablers, and policy influences associated with providing equitable healthy food access in Baltimore, MD, carryout restaurants. We found that the menu offerings in carryout restaurants are determined by individual-level factors, such as restaurant owners’ perceptions of what customers like and owners’ cultural identities; interpersonal-level factors, such as the relationship between carryouts and customers; and institutional-level factors, such as food costs. At the policy level, difficulties in accessing funding information and constant regulation of carryout restaurants strongly affected the owners’ bottom lines. All these factors are intertwined.

Owners’ opinions in this study were divided on whether customers are interested in healthy menu offerings, reflecting a broader issue observed in the literature. Customer demand was identified as both an enabler and a constraint in a scoping review of restaurant interventions [13]. The mixed perspective on customer preferences can be attributed to several factors. Firstly, economic constraints are a significant determinant of food choices in low-income neighborhoods, such as those in HFPAs in Baltimore. Many restaurant owners, both in this study and others, have voiced that customers often prioritize affordability in their food decisions, leading them to opt for cheaper, energy-dense foods rather than healthier options, which are perceived as more expensive [14,15]. Beyond food cost, people’s food preferences are also influenced by their social networks, nutrition knowledge, media influence, work demands, and lifestyle choices [30]. Overall, interventions using pricing strategies suggest that lowering prices could increase purchases of healthy foods [13,31]. This indicates that if restaurants could source healthier ingredients at lower costs, use alternative cooking methods (e.g., grilling versus frying), or receive subsidies for offering healthier prepared food at cheaper prices, it might increase the promotion of healthier menu options. Secondly, cultural preferences and habitual eating patterns also play a critical role in food choices. In communities with long-standing dietary traditions, there may be resistance to changing established eating habits, such as consuming large portions and meat-heavy meals, even in the face of diet-related diseases [32]. Some owners may interpret this resistance as a lack of interest in healthy foods, when in fact, it could be a reluctance to deviate from tradition and cultural values. This can create a cycle where the perceived lack of demand discourages owners from offering healthier options, which in turn limits customer exposure to such options.

Owners’ motivation to maximize profits has been identified as a barrier. Although several owners in this study, as well as in other research, have claimed to be committed to serving their communities [13], the need to “survive” often leads carryout owners to prioritize expenses over the potential revenue that new healthy menu items might generate. Other studies have found that many restaurant owners, both in fast-food and casual dining, prioritize maximizing their sales and profits over the healthy food offerings. Profit is the primary consideration when making changes to their menu [15].

We discovered that macro forces such as regulations, policies, and taxation affect independent carryout restaurants by influencing their economic viability, potentially creating barriers to offer healthy food. As components of a broader economic ecosystem, small businesses often find themselves disproportionately affected by policies that may either overlook their circumstances or impose undue burdens due to their limited financial resources. For example, the foam ban implemented by Baltimore City in 2019 inadvertently burdened carryout owners with increased costs for alternative food package materials [27], which owners indicated had significant financial repercussions. In contrast, it is plausible to anticipate that such policies may have a lesser impact on chain or upscale restaurants, with research suggesting that these larger establishments may even reap financial benefits from “green” initiatives like the foam ban in the long term [33]. Moreover, small restaurant owners highlighted the challenge of operating within narrow net profit margins, particularly when revenue is consistently allocated to taxes and operating fees. This suggests that government support, such as creating grants for small businesses to support sustainability initiatives and offset operational costs, could mitigate this issue [34]. Despite these findings, we observe that much of the existing research on the intersection of policy and healthy food availability in restaurants focuses on nutritional guidelines [35,36], rather than the larger systems impacting small businesses’ ability to operate and procure healthy foods. Thus, there is a pressing need for a comprehensive exploration of policy impacts on small, independent restaurants within a systemic framework.

Carryout owners also revealed financial inequities, especially in uneven neighborhood investments. These disparities stem from structural racism policies like redlining, which restricted resources for communities of color, including food access [37]. Many of the carryouts we interviewed served fried chicken, reflecting the prevalence of “chicken box stores” in Baltimore. These unhealthy food outlets are driven by a perceived supply and demand for unhealthy foods within the community [38]. Since all our participants were recruited from neighborhoods within HFPAs, it is not surprising that discussions frequently touched on the unhealthy food supply and demand and the impact of neighborhood underinvestment. Issues such as higher crime rates, vacant houses, and poor cleanliness deter patronage and negatively affect the community food system [39]. One possible solution includes mechanisms like universal basic income (UBI). A scoping review of the effects of basic income-like interventions in high-income countries found that such interventions led to improved nutrition, reduced property crime, and reduced financial strain [40]. Implementing UBI could provide financial stability for residents, enabling them to make healthier food choices and contribute to a more equitable food system.

When asked about food policy, food assistance programs like the SNAP and RMP, which benefit community members, were discussed during interviews and CAB meetings. Restaurant owners expressed confusion or lack of awareness regarding these programs, often using the terms interchangeably and being uncertain about the application process. Despite the RMP being established in Maryland four years ago, this indicates inadequate communication from program implementers. Research suggests this issue is widespread among small businesses in the U.S. due to minimal marketing efforts [41], despite these programs’ potential to improve nutritional outcomes [42,43]. To enhance healthy food choices and support local businesses, marketing efforts for the RMP in Maryland should focus on independent restaurants. This could increase access to healthier food options and stimulate the circulation of SNAP dollars throughout the local economy [44].

Administrative burdens underlie the challenges of being a carryout owner, constraining their ability to provide community benefits through social programs, grants, or basic operational certificates. These challenges are further exacerbated for immigrant carryout owners, who have increasingly become the business owners in Baltimore [21]. These burdens include language barriers, complex documentation, and fees, which can hinder access to basic public services [45]. Immigrant carryout owners expressed frustration over possible mistreatment from bureaucracies due to their status. One owner with limited English proficiency, who self-identified as undocumented, faced difficulties obtaining food handler licenses, necessitating travel to another state for translated documents. Limited English proficiency also leads to miscommunication [46], overlooking customers’ healthfulness requirements in purchasing food. Furthermore, immigrant owners also voiced that they struggled more with obtaining necessary business ownership documentation, such as Individual Taxpayer Identification Number (ITIN), Tax ID, and Employer Identification Number (EIN) compared to their U.S. citizen counterparts.

### 4.1. Implications for Policy

The findings from this study highlight several critical implications for policy aimed at improving healthy food access in urban, under-resourced neighborhoods. First, the use of subsidies to incentivize the provision of healthier prepared foods at lower prices could address financial disparities and promote equitable resource allocation across different communities. Second, implementing UBI may ensure that resources and investments are equitably distributed among individuals, households, neighborhoods, and communities, thereby providing residents with the financial stability needed to make healthier food choices. Third, food assistance programs such as the SNAP and RMP should increase their marketing efforts and encourage the participation of independent restaurants. This would allow small businesses to benefit from government funding, while also expanding access to healthier food options for low-income populations. By addressing these areas, policymakers can support small, independent carryout restaurants in their efforts to offer healthier menu options and create a more equitable food system.

### 4.2. Limitations and Strengths

To our knowledge, this study is the first to examine structural policy drivers and their relation to barriers and enablers for providing healthy food options in the context of carryouts in urban areas with limited healthy food access and high poverty rates. Our study successfully interviewed a variety of restaurant owners, including those newer to business and those inheriting family businesses. Though we have a small sample size due to recruitment challenges (owners with busy schedules, not present, or not wanting to participate), our interviews reached data saturation. We captured both the operational and personal experiences of running a carryout restaurant business, exploring the narrative intersections of sociocultural, business, and policy factors that shape the carryout food system. Additionally, we obtained geographic variety within the city (east vs. west), enabling discussions on investments and disinvestments surrounding small-business-like carryouts and their communities.

Our findings are robust due to data saturation. Many concerns voiced by carryout owners related to food offerings are interconnected. While studies show customers want healthier foods, many owners do not believe this [47]. At the same time, owners enter the business to make a profit, but as inflation, taxes, and general operating costs increase, they need to raise the prices of their offerings, ultimately impacting customers with budget constraints. This results in less revenue for owners to afford healthier, usually more expensive foods, creating a cycle that hampers restaurant operations. This is especially dire in this study since the establishments are already located in areas with minimal healthy food access. However, if restaurant owners lack access to evidence, current practices will prevail. This highlights the need for clearer, more accessible research, programs, and policies to help owners improve offerings and understand customer demand.

However, several important limitations arose. First, the small sample size may not provide a fully comprehensive view, and future studies with larger samples are necessary to generalize these findings more broadly. Second, we were only able to interview carryouts from eight out of the intended thirteen zip codes, with some participating carryouts in the same zip code. In the five zip codes that we were unable to recruit carryouts, we noticed several limiting factors that prevented us from engaging with carryout owners. These zip codes have historically higher poverty rates and marginalization, potentially causing distrust of researchers from perceived “privileged” institutions. Furthermore, some carryouts mentioned time constraints, and we could only interview owners fluent in English.

This study aimed to capture potential barriers and enablers for ensuring equitable access to nutritious food in Baltimore’s carryout food system from the perspectives of carryout restaurant owners. We did not obtain perspectives from customers nor ask about the carryout owner’s race and identity, which warrant improved study design and further studies.

## 5. Conclusions

This qualitative paper discusses the structural factors affecting healthy food access in small, independently owned carryout restaurants located in urban, under-resourced neighborhoods from the stakeholders’ perspectives. These factors are multi-leveled and multifaceted, spanning intrapersonal, interpersonal, sociocultural, community, and policy domains. Future programs and interventions should address the identified barriers and leverage existing community strengths. Nutrition assistance programs, such as the RMP and SNAP, should increase outreach to business operators and provide application guidance within low-income communities to eventually reach eligible participants. The study also reveals a communication gap between policymakers and small business operators regarding support for sustaining small businesses in these areas. Researchers and public health professionals should work to ensure that messages from this underrepresented population effectively reach policymakers.

## Figures and Tables

**Table 1 nutrients-16-03028-t001:** Characteristics of thirteen carryout restaurants in HFPAs in Baltimore City, Maryland.

	Characteristics	n (%)
Carryout Level	Type of Cuisine	
	Italian	3 (23)
	Seafood	3 (23)
	General/Mixed/American	2 (15)
	Chicken Meals	2 (15)
	Soul Food	1 (8)
	Caribbean	1 (8)
	Asian	1 (8)
Neighborhood or Zip Code Level ^1^	Average HFAI	
	>9.5	5 (38)
	≤9.5	8 (62)
	Median Household Income (in dollars)	
	>40,000	9 (69)
	≤40,000	4 (31)
	Household without Vehicles Available (%)	
	>30%	13 (100)
	≤30%	0
	Total Amount Invested in Small Businesses per 50 Businesses (in dollars)	
	>1,000,000	4 (31)
	750,000–1,000,000	4 (31)
	<750,000	5 (38)

^1^ Data on median household income and the percentage of households without vehicles are sourced from the U.S. Census Bureau and reported at the zip code level. The average HFAI and total amount invested in small businesses per 50 businesses are sourced from the Baltimore Neighborhood Indicators Alliance and reported at the community statistical area/neighborhood level.

**Table 2 nutrients-16-03028-t002:** Supporting quotes for interpersonal drivers.

Themes	Sub-Themes	Supporting Examples
Interpersonal Drivers	Perceptions of consumer preferences	1. “’Cause majority the perception where the assumption is basically, Oh, city people don’t like healthy stuff. But that’s, that’s not just true. Because they love. You have to give them the same quality that you would give them in the county or somewhere that is-let’s say people are more fortunate or have a little bit more money.” 2. “And this is just something that’s just started happening. They’ll ask, if we have any vegetarian options and I’m like, ‘Well, right now, the best that I can do-We do serve some vegetables, but as far as a particular vegetarian dish. We haven’t really looked into that as of yet. Maybe in the future.’” 3. “I have no idea how to make that healthy. Just the vegan, I guess.” 4. “Cheap, hot, fast. I mean, it’s not the healthiest thing in the world, but people are on a certain budget. Sometimes they don’t want to spend $18 for a plate over here or somewhere.” 5. “A lot of people say they like their grits and the breakfast sandwiches. So I think if we tried to change them up, it could go either good or bad. I’m not really sure, but I think that people like them the way it is now.” 6. “I think the most challenging part is the language barrier, kind of. My mom is pretty… She’s not fluent at all in English. And so it’s sometimes pretty hard for her to understand the customers, kind of have the customers understand her.”
	Care for consumer well-being	7. “The old menu, it seemed to me it’s like a less quality type of cuisine, cheaper brands. So, what I’m doing is I’m trying to get better quality stuff and drop the prices. So right now, this area, if you only have $4, you can only get pretty much a cheeseburger, a hamburger and it’s all greasy stuff. So now what I’m trying to offer is a side salad with tuna salad on the side. Offer more vegetables, offer more things they haven’t tasted as far as spices and stuff like that.” 8. “Some people want clubs. We didn’t have them before when we first opened. They want cold cuts, turkey, not red meat. And you have people that do not eat pork bacon. We have turkey bacon because of religion or they just don’t want it.” 9. “Well, we have a lot of customers, like people with disabilities, they come sometimes and we say, ‘We don’t need anything.’ And some days, we prepare them food and we give it to them.” 10. “And I tell you, in my store nobody goes hungry. My cashiers and everybody know that it doesn’t matter how they look, if they look poor or they look rich. If they come to you, they’re hungry. You give them, they eat something. So that’s something they’ve been telling me that goes a long way.” 11. “Somebody found a [bay] leaf in the rice and the person thought this was something really nasty and he was really angry with my employee on the phone. I knew exactly what it was, I said, ‘can you bring him back and I’ll replace it for you?’ So when he came back and I opened my spices that I put in the rice and I pulled it out and I said, ‘this is what it is’. I said, this is all-natural stuff that I put in rice. That’s what I make my rice taste better. I just don’t give you boil rice or I don’t just slap the rice there and give it to you.”
	Maximizing transactions	12. “And that all stemmed from basically making so much, such a great profit off this place. It was extremely lucrative. However, I do not know what pushed him into the realm of just going strictly towards selling seafood. But that’s not all he did. Like I said, he owned businesses like buildings where he rented out the businesses. So he kind of did like real estate.” 13. “I mean, naturally Baltimore has a bad rap period, so I don’t go anywhere else in Baltimore. I go from here and home.” 14. “So, it kind of goes down to the manager we had over in the old market. She was a vegan and she would always ask me, and I’m like, ‘I cannot purchase these things in this market ’cause there’s no demand for it over there.’ But over here there is.” 15. “I did business the wrong way. I was treating customers… I was almost racist.”

**Table 3 nutrients-16-03028-t003:** Supporting quotes for sociocultural drivers.

Themes	Sub-Themes	Supporting Examples
Sociocultural Drivers	Multicultural Ownership Shapes Food Offerings	“It’s gonna be fast food, Chinese food, we have three, and also the Latin food.” “We have homemade spaghetti, homemade meatballs.” “And just being within that neighborhood, it’s just your typical carryout, the Chinese food, subs, sandwiches. So he just wanted to bring something different. So he decided to do a soul food type, something home-cooked.” “It’s exciting to them too. Because, I mean, they were customers getting pizza, pizza, pizza. Now they come and they’re getting food, they’re getting rice, they’re getting something solid.” “I don’t want to do a drastic change, because they are accustomed to the finger food, so we want to take it gradually. So, I’ll be slowly starting to introduce the jerk chicken, the peas and rice, the curry chicken, and they love that. So, I want to- as I build the clientele with that, I’ll kind of expand the menu to oxtail and… all of the different Caribbean foods.” “I started about six, seven years with the healthy stuff that they cannot get in the city. They have to go far away to get the healthy stuff like chicken oil rice. I do many things like steak oil rice and fish oil rice, those are grilled items. I do not use a lot of spice. I use all natural ingredients, yogurt is my base. And then I put just a little salt, black pepper and garlic. I marinate my meat and it goes on the grill, and I don’t fry them. Even the rice I cook, I put a very little bit of oil. I make sure it is done because I eat… Rice is very big in Asia. I know how to cook it. So I cook them really healthy way and then I put lettuce, a salad like the Romanie lettuce with tomatoes, and then put the rice on the side, and put the meat on the rice.”

**Table 4 nutrients-16-03028-t004:** Supporting quotes for business drivers.

Themes	Sub-Themes	Supporting Examples
Business Drivers	Cost of business impacts carryouts’ ability to offer healthy options	“So whatever is cheaper, they give us, so we survive. Because our small business- the rent, taxes, and employees, so if it’s too high, we cannot survive. First day before ordering, we call and check the price. How much is the price for French fries? He said $33 for a box. Other said $32, so we get $32. Because it’s a penny, a penny we make it. We have homemade spaghetti, homemade meatballs.” “It can kind of be challenging sometimes because the prices go up all the time.” “So, I might go out and say, ‘Okay, I want to do this special today, this special, this special,’ and then when I go to the Restaurant Depot, I’ll get the stuff. And then some people, ‘Oh, well, I don’t like this, I don’t like that’. So we try to incorporate it and have as many substitutions as possible, because everyone does not have the same favor. That’s really one of the big challenges is—making sure that we have a substitution for something.” “This is the same food. But taste is different. And the equipment is new coming, so change. No too much oil. So people like it…Taste is a yummy taste, old people easily eat.” “Because this type of equipment is like cookers. If you boil in a cooker and you boil in another pot, there’s a difference. If you cook in a machine, it tastes, and you can easily eat. We—maybe two months after, because the equipment is very expensive. 14,000 for one. I need three equipment, so adds up to 42,000. So, one by one… I already bought one. And after two, three months, I will buy one more. Then we start the new menu.” “My gas, electric. The same gas and electric I used as I did last year, at least $400 more a month. So I have to raise my prices to cover that $400.”

**Table 5 nutrients-16-03028-t005:** Supporting quotes for policy drivers.

Themes	Sub-Themes	Supporting Examples
Policy Drivers	Taxation	1. “Because our small business- the rent, taxes, and employees, so if it’s too high, we cannot survive.” 2. “So they [Customers] gotta understand that it’s not us doing it, but it’s the Maryland sales tax. Then a lot of people complain, a lot of people complain about it… But if you go anywhere, no matter where you go, you have to pay Maryland’s sales tax.”
	Government regulations and policy	3. “Each carryout is not the same. Say if we wanted to do fountain drinks, you have to have a certain permit for fountain drinks. You gotta have a certain permit if you wanna do alcohol…you gotta have a certain permit if you wanna, actually have a sit-down spot.” 4. “We make sure we keep everything up to date. ’Cause they [Health Department] definitely do get on you if you don’t have your right licenses and whatnot. And, the city does come in periodically and, they do checks and if stuff is not right, they will shut you down because we do know of a few places that have been, shut down because stuff wasn’t in order.” 5. “Instead of paying $0.03 for a [styrofoam] box, you’re paying a $1.03 for this eco-friendly platter box. So yeah, over the past couple of years, they’ve [Maryland] really hit carryouts and restaurants hard with all these regulations and bag bans… there’s no point.” 6. “We wait a whole year just to open because waiting on this bench inspection, waiting on the city [Health Department] to come, it was a challenge.” 7. “It’s gonna be the fire department inspection, then… a city inspection, the building inspection.” 8. “We had to call and make an appointment, and then they call and cancel. And then we have to make the appointment again.” 9. “If we went to add something [on the menu], we had to pay extra and we had to call the city and let them know, and they had to come and check everything we’re gonna do, so then they gonna say, ‘Yes, you can put it or not.’” 10. “It’s [Carryouts] not like chains, where there’s a lot of funding. There’s a lot of room for mistakes, room for leeway. So, I think in terms of that, we do need a little bit more attention so that we can actually keep afloat and manage.”
	Information Challenges	11. “So it’s stuff [grants] out there, but you really have to do your groundwork really, and it’s not as easy as people think it is to get some of these grants for minorities or whatnot.” 12. “I’m not so sure because I’m in the process of the whole thing, [SNAP] application, new application and everything, but I think you have to have a certain amount of items in the store and carry… You must carry certain items in the store.” 13. “We had to wait five months just to get an IRS [EIN] number, it’s like the social security number for the place… a USA or American citizen, they gonna get [it] in the next day…it was a different process because we got something called ITIN number, it’s like an identification number. And they say, ‘Oh, if you come with a social security number, it’s gonna be different.’” 14. “We had to get the manager food licenses, and we had to go to Virginia because there is no classes here [Baltimore] on my language.”
	Uneven investment across neighborhoods	15. “Once they [policymakers] get their juice, that’s all they care about. And I really think that they should stop and think about people sometimes.” 16. “[There have been] a myriad of businesses that’ve been in and out of this whole block here… I think it’d be upsetting if this place ever closed because of the city not helping us out more… I think the mayor needs to help support mom and pop establishments.” 17. “Downtown Baltimore does need a little bit more support and help from the government. ’Cause obviously it’s very impoverished very… The literacy rates are low. It’s very… It’s just getting by, you know?… I think whether it be homes, just communities and the businesses that are in this area, I think it’d be nice to get a little bit more support from the government in that kind of aspect.” 18. “I think almost on a daily basis there is a lot of people on drugs, a lot of homeless people that come in, which kind of disrupts the environment for a lot of our other customers as well. And it’s just overall kind of hard to deal with stuff like that.” 19. “The ones [Carryouts] that really don’t care about the business or their neighborhood, they’ll let drug dealing in their store, they’re going to let anything happen in their store.”

## Data Availability

All data generated or analyzed during this study are included in this published article. The raw data supporting the conclusions of this article will be made available by the authors on request due to privacy reasons.

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
