# Peer review of "Barriers and Enablers for Equitable Healthy Food Access in Baltimore Carryout Restaurants: A Qualitative Study in Healthy Food Priority Areas"

_nutrients, 2024, doi:10.3390/nu16173028_

Round 1

Reviewer 1 Report

Comments and Suggestions for Authors

Hua and colleagues present a critical and timely issue: the barriers and enablers to promoting equitable healthy food access in under-resourced, historically marginalized neighborhoods in the U.S.

The study addresses an important public health issue—access to healthy food in historically redlined Black neighborhoods. Conducting in-depth interviews with restaurant owners is a sound methodological choice for exploring the nuanced factors that influence food access in these communities. 

First, I believe that the title is too big and should be changed.

Also, the study is based on only 13 interviews, which may not provide a comprehensive view of the challenges and opportunities faced by small, independently owned carryout restaurants in these neighborhoods. While qualitative research does not aim for generalizability, the small sample size should be acknowledged as an important limitation.

The article mentions that neighborhoods were "purposively sampled" within Healthy Food Priority Areas (HFPAs), but it does not explain the criteria for this selection.

The mixed perspectives on customer preferences for healthy food are mentioned, but there is no real important discussion of why these perspectives differ or how they impact the feasibility of promoting healthier options.

Moreover, specific policy recommendations would strengthen the text by offering actionable insights.

Furthermore, this text should not be a missed opportunity for broader implications, and the discussion should delve into them.

Author Response

Comments and Suggestions for Authors

Hua and colleagues present a critical and timely issue: the barriers and enablers to promoting equitable healthy food access in under-resourced, historically marginalized neighborhoods in the U.S.

The study addresses an important public health issue—access to healthy food in historically redlined Black neighborhoods. Conducting in-depth interviews with restaurant owners is a sound methodological choice for exploring the nuanced factors that influence food access in these communities. 

First, I believe that the title is too big and should be changed.

We appreciate the suggestion regarding the title. We propose changing the title to: "Barriers and Enablers for Equitable Healthy Food Access in Baltimore Carryout Restaurants: A Qualitative Study in Healthy Food Priority Areas." (Ln1-4)

Also, the study is based on only 13 interviews, which may not provide a comprehensive view of the challenges and opportunities faced by small, independently owned carryout restaurants in these neighborhoods. While qualitative research does not aim for generalizability, the small sample size should be acknowledged as an important limitation.

We acknowledge the small sample size as a significant limitation of our study. We added a sentence to the limitations section of the discussion, acknowledging that our small sample size limits the generalizability of our findings, and that future studies with larger samples are warranted to validate and broaden the applicability of these results. (Ln537-539)

The article mentions that neighborhoods were "purposively sampled" within Healthy Food Priority Areas (HFPAs), but it does not explain the criteria for this selection.

We agree that the sampling criteria need further clarification. We have revised Section 2.2 to include a more detailed explanation of the criteria used for selecting zipcodes and carryout restaurants. (Ln129-137)

The mixed perspectives on customer preferences for healthy food are mentioned, but there is no real important discussion of why these perspectives differ or how they impact the feasibility of promoting healthier options.

We appreciate the reviewer’s feedback on the need for a more detailed discussion of the reasons behind the mixed perspectives on customer preferences for healthy food and their impact on promoting healthier options. In response, we have revised the second paragraph in the discussion section to explore these factors more comprehensively, including economic constraints, cultural preferences, habitual eating patterns, and other relevant influences. (Ln397-428)

Moreover, specific policy recommendations would strengthen the text by offering actionable insights.

We appreciate the reviewer’s feedback regarding the need for specific policy recommendations. We have already mentioned these in separate paragraphs throughout the discussion section. Specifically, we proposed: (1) using subsidies to incentivize the provision of healthier prepared food at lower prices, which addresses financial investment and resource allocation across different communities and neighborhoods; (2) implementing universal basic income to ensure resources and investments are equitably distributed among individuals, households, neighborhoods, and communities; and (3) increasing marketing efforts for food assistance programs like SNAP and RMP to encourage independent restaurants to enroll, allowing small independent businesses to benefit from government funding. To enhance clarity, we have summarized these recommendations in a dedicated paragraph as implications for policy. (Ln499-512)

Furthermore, this text should not be a missed opportunity for broader implications, and the discussion should delve into them.

Same as the previous response.

Reviewer 2 Report

Comments and Suggestions for Authors

I would like to commend the authors for their work on the structural factors affecting healthy food access in independently owned carryout restaurants. This study addresses a significant public health issue and provides valuable insights into the challenges faced by restaurant owners in under-resourced neighborhoods.

Summary

The manuscript presents a qualitative study that explores the structural factors influencing equitable access to healthy food in small, independently owned carryout restaurants. The authors interviewed carryout restaurant owners to identify barriers and enablers related to food access in under-resourced neighborhoods. The findings highlight the challenges faced by these establishments and provide insights into potential policy recommendations.

Detailed Comments

1. Abstract

The abstract effectively summarizes the study's objectives, methods, key findings, and implications. However, it could be improved by providing a clearer statement of the main findings and their significance. To provide a more comprehensive overview of the study's contributions, authors could revise the abstract to include a sentence like: "Our findings indicate that financial constraints and lack of awareness about food assistance programs significantly hinder the ability of carryout restaurants in under-resourced neighborhoods to offer healthy food options, underscoring the need for targeted policy interventions to support these businesses and improve community health outcomes."

The list of keywords provided in the manuscript is excessive. The authors should consider consolidating the keywords to focus on the most critical terms that encapsulate the essence of the study.

2. Introduction

The introduction provides a solid background on the topic and clearly outlines the research questions. It effectively contextualizes the study within the existing literature on food access and health disparities.

3. Methods

The methodology is well-articulated, detailing the recruitment process, interview structure, and data analysis methods. The use of semi-structured interviews is appropriate for the research aims. However, the authors should provide more detail on determining data saturation. A paragraph might be included: "Data saturation was determined through ….."

4. Results

The results section presents the findings in a clear and organized manner, with appropriate use of quotes from participants to support the themes identified.

5. Discussion

The discussion effectively interprets the results and connects them to broader implications for policy and practice. The authors provide valuable insights into the challenges faced by carryout restaurant owners. However, the discussion could be enhanced. While the discussion mentions that the study has a small sample size due to recruitment challenges, it does not delve into how this limitation might affect the generalizability of the findings. A more thorough discussion could include potential biases introduced by the sample size and how it may not fully represent the diversity of carryout restaurant owners in the targeted neighborhoods. It would be beneficial to clarify whether the saturation was sufficient to ensure that all relevant themes were captured or if there were still gaps in understanding certain perspectives.  

6. Conclusion

The conclusion summarizes the main findings well and emphasizes the need for policy support for small businesses.

7. References

The reference list is comprehensive and mostly up-to-date. 

Author Response

I would like to commend the authors for their work on the structural factors affecting healthy food access in independently owned carryout restaurants. This study addresses a significant public health issue and provides valuable insights into the challenges faced by restaurant owners in under-resourced neighborhoods.

Summary 

The manuscript presents a qualitative study that explores the structural factors influencing equitable access to healthy food in small, independently owned carryout restaurants. The authors interviewed carryout restaurant owners to identify barriers and enablers related to food access in under-resourced neighborhoods. The findings highlight the challenges faced by these establishments and provide insights into potential policy recommendations.

Detailed Comments

  1. Abstract

The abstract effectively summarizes the study's objectives, methods, key findings, and implications. However, it could be improved by providing a clearer statement of the main findings and their significance. To provide a more comprehensive overview of the study's contributions, authors could revise the abstract to include a sentence like: "Our findings indicate that financial constraints and lack of awareness about food assistance programs significantly hinder the ability of carryout restaurants in under-resourced neighborhoods to offer healthy food options, underscoring the need for targeted policy interventions to support these businesses and improve community health outcomes."

We appreciate the recommendation of the statement, but we want to focus on the enablers and barriers that contribute to how carryout owners interact with the community through the following drivers: interpersonal, sociocultural, business, and policy. It is not just financial and lack of awareness of food assistance program since we want to capture a holistic view of carryouts and community interactions, which is more complex.   

The list of keywords provided in the manuscript is excessive. The authors should consider consolidating the keywords to focus on the most critical terms that encapsulate the essence of the study.

We agree with the reviewer and kept only 5 keywords: independently owned restaurants; healthy food access; food desert; food policy; nutrition program. (Ln32-33)

  1. Introduction

The introduction provides a solid background on the topic and clearly outlines the research questions. It effectively contextualizes the study within the existing literature on food access and health disparities.

Thank you.

  1. Methods

The methodology is well-articulated, detailing the recruitment process, interview structure, and data analysis methods. The use of semi-structured interviews is appropriate for the research aims. However, the authors should provide more detail on determining data saturation. A paragraph might be included: "Data saturation was determined through ….."

We added the following statement regarding the data saturation determination of no new themes or information emerging from the subsequent interviews. (Ln147-148)

  1. Results

The results section presents the findings in a clear and organized manner, with appropriate use of quotes from participants to support the themes identified. 

Thank you.

  1. Discussion

The discussion effectively interprets the results and connects them to broader implications for policy and practice. The authors provide valuable insights into the challenges faced by carryout restaurant owners. However, the discussion could be enhanced. While the discussion mentions that the study has a small sample size due to recruitment challenges, it does not delve into how this limitation might affect the generalizability of the findings. A more thorough discussion could include potential biases introduced by the sample size and how it may not fully represent the diversity of carryout restaurant owners in the targeted neighborhoods. It would be beneficial to clarify whether the saturation was sufficient to ensure that all relevant themes were captured or if there were still gaps in understanding certain perspectives.  

We acknowledge the small sample size as a significant limitation of our study. We added a sentence to the limitations section of the discussion, acknowledging that our small sample size limits the generalizability of our findings, and that future studies with larger samples are warranted to validate and broaden the applicability of these results. (Ln536-538)

  1. Conclusion

The conclusion summarizes the main findings well and emphasizes the need for policy support for small businesses. 

Thank you.

  1. References

The reference list is comprehensive and mostly up-to-date. 

Thank you.

Reviewer 3 Report

Comments and Suggestions for Authors

The study provides important information and is very original.

Here are some recommendations and questions to improve the text.

The abstract would be better in an unstructured format (remove the titles "Background", "Methods", etc.), leaving only the text.

Line 38 - You can use the % symbol

Lines 84-88 - It is better to include the research questions in the form of continuous text.

Line 130 - Wouldn't an appendix with the instrument used fit? This way we will know which guiding questions were used.

Line 149 - Did you have any loss in the sample? If so, it is important to describe it; if not, write that all those contacted responded.

Table 1 - the "N" should be written in lowercase letters.

I believe that visual resources such as word clouds can be valuable even in a qualitative study. This way you can list key words about the terms most frequently used by the interviewees.

Author Response

The study provides important information and is very original.

Here are some recommendations and questions to improve the text.

The abstract would be better in an unstructured format (remove the titles "Background", "Methods", etc.), leaving only the text.

We revised the abstract to an unstructured format, removing section headers to present the content in continuous text. (Ln13-31)

Line 38 - You can use the % symbol

We replaced the word "percent" with the "%" symbol. (Ln37-38)

Lines 84-88 - It is better to include the research questions in the form of continuous text.

We have revised the text to present the two research questions in a continuous format. (Ln97-100)

Line 130 - Wouldn't an appendix with the instrument used fit? This way we will know which guiding questions were used.

We have added the in-depth interview guide as supplementary material and will upload it with the revised manuscript. (Ln161)

Line 149 - Did you have any loss in the sample? If so, it is important to describe it; if not, write that all those contacted responded.

We had a small loss to follow-up of 3 restaurants who expressed interest, but did not schedule an interview. In Lns 148-154, we have added a more detailed explanation of recruitment attempts.

Table 1 - the "N" should be written in lowercase letters.

We have revised "N" to lowercase “n” in Table 1.

I believe that visual resources such as word clouds can be valuable even in a qualitative study. This way you can list key words about the terms most frequently used by the interviewees.

We appreciate the reviewer’s suggestion regarding the use of word clouds as a visual resource in qualitative studies. While word clouds can effectively highlight the most frequently used terms by interviewees, we believe that they are more suitable for presentations at conferences or community workshops rather than for publication in a scientific journal. To enhance the visual appeal of our manuscript and ensure it aligns with the expectations of a scientific publication, we have chosen to include a graphical abstract instead. This graphical abstract succinctly captures the key themes and findings of our study, offering a visually engaging summary for readers.

Round 2

Reviewer 1 Report

Comments and Suggestions for Authors

After some modifications made by the authors, I believe that this paper can be acceptable for publication.